# "I Just Want to Live My Life": Young Disabled People's Possibilities for Achieving Participation and Wellness

**Anna Sigrún Ingimarsdóttir *** and **Snæfrídur Thóra Egilson** 

Centre of Disability Studies, School of Social Sciences, University of Iceland, Sæmundargata 2, 102 Reykjavik, Iceland; sne@hi.is
* Correspondence: asi@hi.is

**Abstract:** This study aimed to (a) explore disabled children's and adolescents' possibilities for participation and (b) identify the practices and policies that affect their participation and how these are enacted. Case studies were conducted with seven children and adolescents with various impairments. Each case included interviews with the young person, their parents and teachers, as well as observations in their usual environments. The interview topics covered the young people's participation, their sense of belonging and aspects that were pivotal to their engagement and wellness. The observations focused on their possibilities for participation and interactions with peers and adults. These young disabled people's possibilities for participation at home, in school and in their neighbourhoods were affected by complex dynamics between personal and environmental factors. Whether and how the young people's disability-related rights were enacted depended on the socio-cultural–material arrangements and parents' knowledge of the welfare system. To better understand and act on the complex and marginalised position of young disabled people, more focus should be directed at policies that affect their rights and possibilities for participation and how these are enacted in practice. Knowledge needs to be expanded to scrutinise the disabling hindrances hidden in social and structural spaces and implemented in services.

**Keywords:** disabled children's rights; socio-material arrangements; services; accommodations; case studies





## 1. Introduction

The significance of promoting and ensuring that disabled people have the opportunities to live well and participate in all areas of society is emphasised in international policy and human rights treaties (Convention on the Rights of the Child (CRC 1989); Convention on the Rights of Persons with Disabilities (CRPD 2007)). The current research has identified various challenges to the opportunities of disabled children and adolescents to participate in social activities and to experience wellness, in comparison with their nondisabled peers (Bedell et al. 2013; Egilson et al. 2017a, 2018; Hamdani et al. 2018, 2022; Imms et al. 2017; Ólafsdóttir et al. 2019). Increasingly, the importance of focusing on ways to promote disabled children's and adolescents' social participation and wellness has been highlighted (Egilson et al. 2017b; Hamdani et al. 2018, 2022; Krieger et al. 2018; Ólafsdóttir et al. 2019; Smith et al. 2021).

Participation is a multi-dimensional construct, with two main dimensions typically identified—an objective dimension reflecting whether someone is included in the routine social activities of a particular setting, how and with whom, and a subjective dimension reflecting a person's engagement, sense of belonging and contentment with one's involvement in that setting (Anaby et al. 2013; Imms et al. 2017; Powrie et al. 2015). In line with the normative understanding that considers disability as being, first and foremost, within the person (Oliver 2013), disabled children's and adolescents' lack of participation has often been regarded as directly linked to their individual impairments. Increasingly, the various

environmental factors that influence these young people's opportunities for participation have been stressed, such as attitudes, the design of built environments, policies, services and systems (Anaby et al. 2013; Egilson et al. 2017a, 2018; Hodge and Runswick-Cole 2013; Ingimarsdóttir et al. 2023; Krieger et al. 2018; Ytterhus et al. 2015). Thus, personal and socio-cultural factors appear to intersect to mediate these children's and adolescents' participation in various situations (Egilson et al. 2021). This resonates with the relational understanding of disability put forward in the preamble of the CRPD (2007), where disability is described as "the result of the interaction between persons with impairments and attitudinal and environmental barriers that hinder their full and effective participation in society on an equal basis with others".

Similar to participation, the concept of wellness can have varied meanings, which nevertheless all emphasise the importance of being able to draw on personal strengths and capabilities in order to live the best life possible (Hamdani et al. 2018, 2022; Rachele et al. 2014). The Global Wellness Institute (n.d.) defines wellness as "the active pursuit of activities, choices and lifestyles that lead to a state of holistic health". How people perceive wellness is considered to be based on their satisfaction relating to the interplay of various dimensions, that is, physical, social, psychological, intellectual, emotional and spiritual (Adams et al. 2000; Global Wellness Institute n.d.). The concept of well-being is often used interchangeably with wellness, although well-being also extends to society's role and emphasises societal responsibility (World Health Organization 2021). In a scoping review focusing on the wellness of young people with developmental disabilities, Hamdani et al. (2018) found that the concept was often ill defined, making it complex to untangle and compare. Overall, disabled children's and adolescents' ideas about wellness underscore their desires to have meaningful opportunities to take part in activities, have a variety of social relationships (Ingimarsdóttir et al. 2023; Jóhannsdóttir et al. 2022; Melbøe and Ytterhus 2017), be respected as individuals (Byhlin and Käcker 2018; Jóhannsdóttir et al. 2022) and have stable medical health (Hamdani et al. 2018, 2022; Rachele et al. 2014).

Ample research demonstrates that these young disabled people's[1] ideas of participation and wellness issues may differ from those of their parents and other adults (Egilson et al. 2017b; Emerson et al. 2023; Hamdani et al. 2019; Hemmingsson et al. 2017; Ólafsdóttir et al. 2019; Silva et al. 2019). Nevertheless, including the perspectives of parents and other key stakeholders may enable the development of a better understanding of a young person's ability to participate and be included in various contexts and to experience wellness (Egilson et al. 2021; Teachman and Gibson 2013).

This research is part of a larger study which focussed on the life quality and participation of disabled children and young people in Iceland (Egilson et al. 2021; www.life-dcy.is). This part of the study aimed to expand prior knowledge on disabled children's and adolescents' possibilities for achieving wellness and participation in valued settings and activities. It also explored how practices, policies, physical layouts, sensory qualities, social relations and attitudes, such as ableist assumptions that favour typical bodies and abilities, shape their opportunities for participation and inclusion. The research questions were the following:

- How do young disabled people perceive their possibilities for participation and wellness at home, in school and in their communities?
- How do socio-cultural–material arrangements, such as policies and practices, intersect to shape these young people's opportunities for participation and inclusion?

### 1.1. Political Environment, Rights and Policies

The CRPD (2007) is a treaty on human rights that reframes disability with respect to human rights and establishes the norm of the societal participation of disabled people on an equal basis with others. The treaty directs policy focus towards human rights and social perspectives on disability, where more attention is paid to the contextual factors shaping disability rather than to individuals and their impairments. The rights to services that support participation in the community, as well as to make decisions and choices in

one's own affairs, are enshrined in the CRPD (Devi et al. 2011; Löve et al. 2018). By signing and ratifying the CRPD, authorities have recognised the importance of their obligations to prevent discrimination and provide opportunities for disabled people of all ages to be active members of society.

Article 7 of the CRPD (2007) focuses on disabled children and emphasises that all necessary measures must be taken to ascertain their full enjoyment of all human rights, including facilitating social inclusion. It similarly states that it must be ensured that disabled children "have the right to express their views freely on all matters affecting them, their views being given due weight in accordance with their age and maturity, on an equal basis with other children, and to be provided with disability and age-appropriate assistance to realize that right". These emphases are consistent with and further outline the provisions of Article 23 of the CRC (1989), which focuses on disabled children's rights to special care and support to ensure that they can live full and independent lives. Furthermore, Article 31 of the CRC highlights all children's right to participate in leisure and recreational activities appropriate for their age, as well as the importance of access to equal opportunities.

Achieving equality for young disabled people is often a matter of having access to the necessary accommodations that may enable them to participate fully in all aspects of life (Navarro 2014). However, reasonable accommodation, as defined in Article 2 of the CRPD (2007), should not "impose a disproportionate or undue burden" on other social actors. Grue (2023) points out that this restriction creates what appears to be an underlying contradiction in the convention. While the right to full and equal participation is nominally absolute, in practice, it depends on a heavily qualified right to reasonable accommodation. Consequently, this principle of "undue burdens" may result in important accommodations not being provided in practice (Grue 2023, p. 2).

*1.2. The Icelandic Context*

As one of the Nordic countries, Iceland has received international attention for its high standard of living, emphasis on human rights in policies and legislation, extensive public welfare provisions and gender equality (Kangas and Kvist 2018). The responsibility for most disability services was transferred from the Icelandic state to municipalities in 2011 to create more cohesive services offered closer to users (Lög um breytingu á lögum nr 59/1992, um málefni fatlaðra, með síðari breytingum, No. 152/2010). Primary schools and lower secondary schools are also funded and administered by municipalities. An approved diagnosis from the authorised organisations is typically required to be eligible for specialised services and support inside and outside of school (Lög um samþættingu þjónustu í þágu farsældar barna 86/2021).

In 2016, Icelandic authorities ratified the CRPD (Stjórnarráð Íslands [Government Offices of Iceland] n.d.), indicating their commitment to implementing the treaty's obligations. Subsequently, the legal framework started to integrate a relational understanding of disability, along with the goals and requirements outlined in the CRPD. Although legal texts and policies increasingly proclaim disabled people's rights to an accessible environment and necessary accommodations, the gaps between policy and practice still pose a threat to the societal participation of disabled people, as is evident in recent research (Brennan and Traustadóttir 2020; Jónasdóttir et al. 2020). This relates partly to the lack of awareness about disability issues and the poor financial situation of many Icelandic municipalities, and it has been pointed out that both state and local authorities have underestimated the scope and costs of the transfer of disability services to the municipalities (Ríkisendurskoðun [The Icelandic National Audit Office] 2021). Moreover, the legislative wording and exemptions allow interpretations and implementations that are not in line with disabled people's core rights, as put forward in the CRPD (Jónasdóttir et al. 2020).

Personal assistance (PA) for disabled people is offered in most European countries and has been considered an important tool for empowerment, independence and participation (European Network on Independent Living 2020). Although traditionally designed for adults, in a few countries, such as Norway and Sweden, PA has been extended to disabled

children and their families (Jenhaug and Askheim 2018). In Iceland, PA payments for disabled children and adolescents had been awarded only in exceptional cases, but in 2022, Úrskurðarnefnd velferðarmála [Welfare Appeals Committee] (2022) decided that applications for PA for children under the age of 18 could not be denied on the basis of age.

## 2. The Study

This study is part of a larger project focusing on young disabled people's lives. The design and methods of the larger study are thoroughly described in an earlier publication (Egilson et al. 2021). This part consisted of case studies with seven young disabled people and focused on their wellness, as well as the intersecting environmental effects on their rights and possibilities for participation. Case studies (Creswell and Poth 2017) can promote an understanding of the context in which young disabled people's lived experiences take shape, such as the structure of their daily lives, their aspirations and agency and the role of important actors in their lives. Moreover, this methodology facilitates building trust with participants using a stepwise approach in addressing sensitive issues that may arise.

### 2.1. Participants and Procedures

Altogether, seven young disabled people (aged 11–15), as well as key actors in their lives, participated in the study. Five participants were recruited through informants in the disability sector and two through snowballing. Table 1 summarises the key participants' main characteristics. The majority were diagnosed with autism spectrum disorder (ASD) in addition to another type of impairment. Apart from Helgi, who lived in a rural area, all the participants lived in the greater capital area, where approximately 70% of the Icelandic population lives.

**Table 1.** Key participants.

| Pseudonym | Age (Years) | Gender * | Impairment Type |
|---|---|---|---|
| Saga | 11 | F | Physical |
| Gunnar | 11 | M | Physical, ASD |
| Helgi | 12 | M | ASD, ADHD |
| Gísli | 15 | M | ASD, Neurodevelopmental Syndrome |
| Atli | 15 | M | Physical, ASD |
| Elín | 14 | F | Invisible, ASD |
| Hanna | 15 | F | Invisible, Physical |

* Each participant's gender is self-identified.

The data were gathered in 2018–2020. In line with the case study approach, we drew on multiple sources of information (Creswell and Poth 2017; Yin 2009). Each case included three to five interviews with the young disabled person and at least one parent and one teacher, along with observations and document analyses. The interview guides covered various aspects of the young people's participation in different settings, their wellness, friendships and sense of belonging. Additionally, the young people were encouraged to choose conversation themes that they found important. The interviews lasted about one hour each and the observations lasted from two to six hours.

Typically, we started by interviewing the parents to gather background information that would better enable us to build trust, establish a rapport and ensure that we focused on the topics and issues that were relevant to their children (Rasmussen and Pagsberg 2019; Teachman and Gibson 2013). Then, we met with the young disabled person on several occasions. We told them ahead of time what we would ask about in the interview and how long it would last. The interviews with the young people started with simple, precise and structured questions, followed by more open-ended questions. To avoid jumping to conclusions, we emphasised probing and asked directly about certain events and experiences, such as how they understood some incidents that we had observed. The teachers were typically interviewed last. The observations took place in the participants' homes

and schools and focused on their possibilities for participation and their engagement and interactions with peers and adults within these environments. Observations in recreational settings were also planned, but due to the COVID-19 outbreak, only a few were carried out. The participants also shared documents, such as school assignments, photos and drawings, with us. Approximately six months after the last interview, the young people received accessible summaries to review and discuss with us, which also gave them an opportunity to provide additional comments, either in person or online. About five such comments were received and then included in the data.

### 2.2. Data Analysis

The interviews were recorded with the participants' permission and then transcribed verbatim by the authors or the members of the larger team. Each transcribed interview and set of observation notes was reviewed and reread iteratively to determine its accuracy (Brinkmann and Kvale 2015). The data were then grouped and organised according to the characteristics in ATLAS.ti (Friese 2014; Seidel 1998) using a coding system consistent with the research objectives. The initial analysis was inductive and data-oriented (Alvesson and Skjöldberg 2018). By comparing and contrasting the young people's experiences, we focused on the commonalities and differences regarding their possibilities for achieving participation and wellness. Both authors then re-read the coded transcripts multiple times to capture the conflicting issues in the mechanisms that appeared to either facilitate or restrict the young people's participation and wellness and the ways in which their rights were enacted or otherwise.

### 2.3. Ethics

All participants were informed about the purpose of the study and willingly participated. They received an introduction letter and a written consent form, which informed them about the aim of the study, interviews and observations, as well as their ethical rights. Each consent form was signed by the researcher, the young person and the other participants (parents and teachers) before the interviews and observations took place. An emphasis was placed on trust and anonymity, such as by using pseudonyms for people and places. The study was approved by the Icelandic National Bioethics Committee (VSN-13-081/16-187-V2).

## 3. Findings

To illustrate how the young people who participated in our study perceived their possibilities for participation, we provide short descriptions and larger vignettes (Boxes 1 and 2). All emphasised being able to participate in social activities and everyday events along with their peers, but they often experienced limited opportunities due to physical, attitudinal or structural restrictions. The barriers they encountered intersected according to different contexts, such as inaccessible physical layouts, inflexible systems and ableist attitudes, and typically, a more complex picture arose during adolescence. For example, Helgi lived in a rural area and travelled 40 km to and from school on a school bus. After-school activities and outings with friends required preparation in terms of transport, but due to strict procedures and the lack of funding, the municipality did nothing practically or financially to accommodate him and his family. As Helgi came closer to adolescence, he also became more interested in all kinds of social encounters, but had few options, except social media.

Negative attitudes, along with rigid ideals and traditions, were reflected in how the young people's capabilities were not necessarily given credit in the school environment. Elín was very artistic and resourceful and had learned (on YouTube) different crafts that she proudly showed the first author during their first encounter, and also beat the latter in complicated strategic board games. However, these strengths appeared to be unappreciated in school. Her main teacher was pessimistic about her potential and stressed that her work was not in line with the educational standards. She explained, "We have not found Elín's

strengths, and I do not know if we will." Because of this negative attitude, Elín was unable to develop her artistic talents with support from the school. An important opportunity to build up her strengths to the fullest appeared to have been lost.

Obtaining the right kind of support and services (self-defined) was central in all accounts. Typically, assistants were part of the school staff and controlled by the school, leaving the young people with hardly any voice on the matter. Outside school, Atli, Gísli and Elín had support workers who met with them once a week to do something "fun together like going to the movies or cafés" (Gísli). They liked this arrangement and wanted to meet their support workers more often but claimed that the support was often unreliable, such as when the latter cancelled a meeting or resigned. Elín said, "I get upset when she [the support worker] is not going to make it." Difficulties in reaching the "right support person" and high staff turnover limited the usefulness of this arrangement, thereby reducing the young people's opportunities for social participation. In contrast, Gunnar and Saga had recently received PA services, which greatly increased their opportunities to participate inside and outside of school, along with their peers.

**Box 1.** Receiving the right kind of service: "How the PA contract changed my life".

> Gunnar was a well-spoken young man and a talented student. Due to his physical impairment, Gunnar used a walker and wheelchair interchangeably for travel, depending on the situation. Some areas in the school were quite inaccessible for him, such as a fenced-in court in the schoolyard where his peers commonly gathered during recess. From a young age, Gunnar had assistants in many situations in school, provided mainly on the school's terms. Beyond school hours, limited support was available other than that provided by his family. Although after-school services were offered in his school district, these were mainly intended for and used by children aged 6–9. As the years passed, Gunnar and his family increasingly questioned this arrangement, which made Gunnar feel left out in valued after-school activities. As Gunnar described, "I could not do the same things as my classmates, like go to birthday parties and such."
> A lot changed for the better when Gunnar got a PA contract at the age of 11, which gave him and his parents control over how the support was provided, when and by whom. He vividly described these changes in a presentation, which he named "How the PA contract changed my life". Consequently, Gunnar was able to take part on his own terms, making him feel much more independent, self-sufficient and happy. He said, "Now I can do the same things as my classmates, although I may do them a bit differently." Gunnar received PA every day except Sundays, focusing on helping him go home after school, attending physiotherapy, engaging in sports activities and visiting friends. Gunnar and his family prioritised hiring "young guys". According to Gunnar, this was because he needed help with personal issues, such as "going to the restroom", and that they "understood him" and thus could talk to him "about personal stuff". Gunnar was quite comfortable with the young guys joining him to meet his peers in situations where adults were usually not visible. His mother Tinna added, "At this age, you don't necessarily want your mother around." Gunnar claimed that he felt "in control", not only because he could now participate in the way he wanted but also because his parents did not have to help him out during the daytime. He said, "My parents have important jobs, and I want them to be able to work without having to worry about who helps me during the day." Evidently, Gunnar was proud of his parents and found it important that they were given time to work and contribute to society. However, receiving PA did not come naturally, as Tinna described, "We were never informed about the possibility of PA assistance; only because I heard about it from somebody did we think about applying for PA on Gunnar's behalf. But when we applied, we were well received."

When the study was conducted, Saga had recently received a PA contract, which provided her with much more control over the way in which she was provided support. Just as in Gunnar's case, Saga's parents had not been informed about the possibility of PA services but heard about this option by chance. The PA services provided Saga with opportunities to be more independent in her peer relations and—as her mother emphasised—take more responsibility for her actions. Her mother continued, "The [assistants] are not the ones who bring her up; that is my job. They help her with things [that] children typically do. And if Saga skips school, I scold her, not the assistant." Saga and her family found the PA services pivotal to her wellness as they provided her with freedom, autonomy and independence at

the important stage between childhood and teenage years. Other participants did not have PA services, but then again, none of them had extensive physical impairments, which seem to be the criteria for Icelandic children to receive PA contracts (personal communication).

The families of the other young disabled people had also often not been informed about their rights and subsequently missed out on the disability-related support that might have increased the children's possibilities for participation and their parents' possibilities for supporting them. This was a source of frustration for many, including Gísli's mother, who claimed to be exhausted because of all the work involved in dealing with the system. She stated, "If only we had been offered more supports, if only I had known (about our rights)!"

**Box 2.** Atli: Opportunities for participation: "I am not going to accept any segregation".

> Similar to Gunnar, Atli was a bright young man, and due to his physical impairment, he used a walker in physically demanding situations. Atli also had an ASD diagnosis and was particularly sensitive to sounds and crowds. Atli found it important to be socially active and included but claimed that he had often been sidelined and not listened to in school. For example, a few years earlier, school officials had decided one-sidedly that due to his impairments, he was better off in a separate classroom, away from his peers. Atli and his family objected. His mother said, "He was in a class with 50 students, and the noise was just too much for him; decreasing the number of students to, say 20, would have been the solution." For his part, Atli claimed that this shift was due to bullying, and he felt that it was unfair that he was the one made to leave: "Instead of the teachers confronting the bullies, I was removed from the class!" Because of this, Atli felt discriminated against and not respected on equal grounds with his peers. He claimed: "I do not have the right to say 'no'; they just take me away when I just want to live my life like everybody else."
> When opportunities for participation were presented and Atli felt respected and included, he also felt that life was good. At the time of this study, he worked at a specialty shop after school and had been invited to be a spokesperson for young disabled people at a few conferences. Having valuable and respected roles improved his sense of wellness because he felt accepted and heard. Being vocal and visible helped him gain confidence, especially getting a platform in his political party, where he was well received. Atli's self-worth increased as he recognised new possibilities for the future and started to plan his political career, already at age 15. He claimed, "My dream is to be a minister."

Differences in the views of young people and their parents on which opportunities for participation to pursue were occasionally observed. Because of health reasons, Hanna could not attend football practice with her schoolmates. This was a major concern for her parents, who feared that Hanna might be lonely and sidelined by her peers. Hanna regarded the situation differently since she had found new friends in school who did not like football. She was also active on social media, where she reached out to a community of young queer people. Contact with the queer community was a momentous experience for Hanna's identity formation as she was figuring out her sexuality, but she had not yet disclosed this part of her life to her parents.

### 4. Discussion

This study explored young disabled people's possibilities for participation and wellness. Two research questions were posed: (1) How do young disabled people perceive their possibilities for participation and wellness at home, in school and in their communities? (2) How do socio-cultural–material arrangements, such as policies and practices, intersect to shape these young people's opportunities for participation and inclusion? The findings are discussed in detail below.

The young people perceived their possibilities for participation and wellness differently, depending on the time and the context. The younger participants focused on the present and underscored having opportunities to be included in social settings and activities, along with their peers. These were often complicated by inaccessible built environments or by limited, or not the "right kind", of support. Being able to participate without the presence of their parents or other authoritative figures opened opportunities to develop and strengthen their relationships with their peers, which became increasingly important

as the years went by. To a certain extent, the younger children's social environment was not much different from that of other children, and they were involved in activities similar to those of their peers the same age. They were nevertheless left out in certain contexts, such as in the schoolyard. Similar to other studies' findings (Andersen and Dolva 2014; Bekken 2017), initially, the younger participants were not very critical of their limited possibilities for participation and appeared to take the status quo for granted. However, when faced with new options, such as PA, they emphasised the positive impacts of these supports on helping them lead better lives and participate along with others. Indeed, perceptions of what is ordinary or "enough" are not created in a vacuum but formed in relation to the social environments in which people are immersed (Gibson 2016), in this case the children's lack of familiarity with other ways of living. Examples of having more choices and flexibility regarding social participation with peers, as well as the importance of not being "burdens" so that their families could participate in the labour market, were shared. Disabled children are in fact often perceived as burdens on their family (McLaughlin et al. 2016; Runswick-Cole 2013), and it can be difficult to form a positive identity and experience wellness under such circumstances.

A more complex picture arose during adolescence, when the young people faced increased educational demands in school, along with their changed experiences, priorities and a wider field of activity. One participant not only had to negotiate her disability but was also figuring out her sexuality; another emphasised making his political mark in the world. Opportunities for young disabled people to explore their possibilities are pivotal, especially if they belong to multiple minority groups, which can make their social position more precarious (Delgado and Staples 2007; Toft et al. 2019). It is well known that adolescence can be an especially difficult time for disabled individuals, who may experience anxiety, struggles and grief when they try to reach normative milestones that are often rigid, inflexible and difficult to obtain (Hamdani and Gibson 2019; Jóhannsdóttir et al. 2022).

In all seven cases, various socio-cultural–material arrangements intersected to shape the young people's opportunities to participate and experience wellness. Many of the hindrances they faced were rooted in policy or service implementation, which did not accommodate their need for support. For example, assistants were typically employed by schools or social services and according to these systems' specific terms. Other barriers included inaccessible built environments and negative attitudes, such as when the young people's non-academic talents were not valued in schools, as well as the way in which the schools dealt with bullying.

Our findings also showed the striking lack of information provided to the young disabled people and their families regarding various disability-related rights. Consequently, a few families had to actively seek out and negotiate support to facilitate their children's participation and wellness. The parents vividly described how they had been unaware of their rights and that service providers did not initiate contact with information regarding services. In some instances, this absence of an incentive led to a lack of important accommodations that could have benefited the young person and their family for years. This echoes the findings from previous research with young disabled people and their families, which has consistently shown a notable shortage of information and the difficulties in accessing support from the systems with which and professionals with whom the families interact (Egilson et al. 2017a, 2018; Tøssebro and Wendelborg 2015). This work of having to find services and support is arguably one form of what Grue (2023, p. 2) calls "undue burdens", referring to the often-invisible work imposed on disabled people and their families to ensure their participation.

Although the parents were the young people's strongest allies and worked hard to provide their children with opportunities for participation and autonomy, there was not always a consensus on how to act or what to prioritise. On some occasions, the parents' stance appeared to be affected by normative thinking, such as when they wanted their children to join peers in particular activities, while the young people had different priorities.

It is well known that young disabled people's ideas on wellness issues tend to differ from those of their parents (Egilson et al. 2017b; Hemmingsson et al. 2017; Ólafsdóttir et al. 2019; Silva et al. 2019) and—with good intentions overruled—ableist ideas may surface (Fine 2019; Goodley et al. 2019; McLaughlin 2023) because of the effects of living in an ableist society.

Although in this paper we have mainly focused on socio-material arrangements from a perspective based on human rights, we also want to point out the significance of the ways in which bodies appear and function regarding whether they are perceived as fitting in. Different kinds of impairments may lead to various forms of prejudice because of normative ideas about how bodies should appear and behave (McLaughlin et al. 2016). Arguably, personal and environmental dimensions uniquely interacted in forming the young people's experiences of participation and wellness. In general, more accommodations were provided in terms of the challenges faced by our study's participants due to visible physical impairments, although, in many cases, they took a long time to be implemented, such as PA services. This is important since without mobility, young disabled people can be excluded from everyday activities that support wellness and engender a sense of belonging (Smith et al. 2021). The participants' less visible impairments and characteristics, such as those due to ASD, were typically met with less understanding, and the interviews with their teachers conveyed that these students were occasionally regarded as a nuisance to a smooth-running classroom. Our survey studies on the participation of young autistic people (Egilson et al. 2017a, 2018; Jakobsdóttir et al. 2015) revealed a confluence of environmental factors that strongly affected their possibilities for participation. Similar findings were identified in a scoping review focusing on environmental aspects that either supported or hindered young autistic people's relationships and participation in social activities (Krieger et al. 2018). This directs attention towards how young disabled people are made to feel out of place through the ways that schools and other settings in which they typically interact are designed and organised, and where conformity is emphasised (McLaughlin et al. 2016).

PA services for persons under 18 are exceptions in Iceland and elsewhere (Jenhaug and Askheim 2018). These PA services' positive impact on increasing participation and wellness is clearly reflected in a participant's description: "How the PA contract changed my life." This related to minimising stigmatisation so that the young person—together with an assistant—could have experiences similar to those of their same-aged peers, such as going to and from school, being more responsible and independent and venturing beyond the family sphere. Similar aspects have been raised in other studies (e.g., Jenhaug and Askheim 2018), although publications on PA for disabled children and their families have typically focused on the effects on the families, not on the benefits for the disabled young people. There is a need for more studies on PA services' impacts on young disabled people.

The findings from our previous studies underscored young disabled people's sense of belonging and acceptance as the cornerstones constituting the goodness of life (Egilson et al. 2021; Ingimarsdóttir et al. 2023; Jóhannsdóttir et al. 2022). Hence, whether and how practices and policies facilitate young disabled people's wellness and participation in valued activities needs to be further scrutinised and acted on.

Given the small sample size, our findings cannot be generalised to the broader population of young disabled people. Moreover, due to the COVID-19 pandemic, we were unable to focus as much on participation in community settings as planned. We also acknowledge our Western perspective and that some of the issues raised in this paper may not apply to other cultures where young disabled people's possibilities for participation and wellness may be even more limited than in Iceland (World Health Organization 2011, 2022).

## 5. Conclusions

The young disabled people and their families worked hard to ensure their participation in valued activities. Access to the relevant accommodations and support provided participation opportunities and promoted a sense of wellness so that the young people felt that they had a place in the world. However, the burden of finding accommodations

was typically imposed on the parents, who had to work extra hard to ensure that their children could participate along with their peers. Translating the CRPD's (2007) human rights perspectives effectively into practice so that policy and service implementation aligns with its principles requires an emphasis on the ways in which important information is provided and the availability of the necessary support when and where needed.

**Author Contributions:** Conceptualization, A.S.I. and S.T.E.; methodology, A.S.I. and S.T.E.; software, A.S.I.; validation, A.S.I. and S.T.E.; formal analysis, A.S.I. and S.T.E.; investigation, A.S.I. and S.T.E.; resources, A.S.I. and S.T.E.; data curation, A.S.I. and S.T.E.; writing—original draft preparation, A.S.I.; writing—review and editing, A.S.I. and S.T.E.; visualization, A.S.I. and S.T.E.; supervision, S.T.E.; project administration, S.T.E.; funding acquisition, S.T.E. and A.S.I. All authors have read and agreed to the published version of the manuscript.

**Funding:** The study was funded by the Icelandic Research Fund, grant number 174299-051 and the University of Iceland Research Fund (2018–2019).

**Institutional Review Board Statement:** The study was conducted in accordance with the Declaration of Helsinki, and approved by the Icelandic National Bioethics Ethics Committee (protocol code VSN-13-081/16-187-V2 date 26 September 2017).

**Informed Consent Statement:** Informed consent was obtained from all participants involved in the study.

**Data Availability Statement:** Data used in this research is unavailable due to privacy.

**Acknowledgments:** The authors would like express heartfelt thanks to the young people, parents and teachers who contributed so generously to this study. Thanks are extended to Ásta Jóhannsdóttir and Linda Björk Ólafsdóttir—our fellow researchers in the LIFE-DCY team—who took part in data gathering.

**Conflicts of Interest:** The authors declare no conflict of interest.

## Notes

[1]    In line with our theoretical standpoint, the term young disabled people is used throughout this article to signify that disability arises in the interaction between young people with impairments and their environments. Young people with impairments may indeed have considerable abilities. Although they may be disabled they do not "have" disabilities.

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
