# Peer review of "“I Just Want to Live My Life”: Young Disabled People’s Possibilities for Achieving Participation and Wellness"

_socsci, doi:10.3390/socsci13010063_

Round 1

Reviewer 1 Report

Comments and Suggestions for Authors

Because of your dual focus on social participation, and the promotion of wellness, your article examining the situation of young people with disabilities has the potential to be of notable interest to the readers of Social Sciences. Your review of the literature has presented key issues about the importance of social participation for young people affected by disabilities, and carefully provides the rationale for studying the ways they can participate in meaningful activities, and promote their own wellness. Although the size of your sample is modest, you have managed to observe and record key patterns that promote social participation of those with disabilities, and lead to greater wellness in their lives. There are some sections that could be revised for greater clarity, and some sections that need further elucidation.

Abstract

Please remove the designation of sections (Aims, Methods, Findings, Conclusions) as they distract from your brief description.

p. 1, line 7. It would be helpful for you to indicate how many children and how many adolescents are in your sample.  

p. 1, line 8. Please indicate what you were observing.

p. 1, line 14.  You discuss the enactment of young people’s rights as due to chance.  In your description, you seem to make the case that there were strong influences that shaped their experiences.

Introduction

p. 1, line 25.  Please revise to state “human rights treaties.”

p. 1, line 35.  Consider changing to “…objective dimension reflecting whether someone is included…”

p. 1 line 40.  Would be clearer if stated as “…disabled children’s and adolescents’ lack of preparation…”

p. 2, lines 64-68.  Please edit this complex statement to provide greater clarity.

p. 2, lines 7i3-75.  Suggest you change to “…including the perspectives of parents and other key stakeholder may enable the development of a better understanding…

p. 2, line 81.  Please define “ableist assumptions” earlier in the introduction.

p. 3, line 101.  It would be helpful for you to clarify your meaning of “mediating social inclusion.”

p. 3, lines 125-131.  Please cite these sources in a briefer way if possible.

p. 3, line 140-141.  What is the specific lack of knowledge you are referring to here?

p. 4, line 154.  Do you mean “applications for PA for children under the age of 18”?

Methods

p. 4, line 160. Please indicate what type of participation is possible.

p. 5 lines 200-201.  Please specify how many comments from how many participants.

p. 5.lines 219-220. Were “other participants” always parents?  Please indicate the roles played by the other signers.

Findings

p. 6, Vignette 1—Really helpful example!

p. 7. Line 311.  Not sure what is meant by “Instead of the teachers fronting the bullies..”  Should it be changed to “confronting”?

Discussion

p. 8. Lines 363-365.  Please consider expanding this section, which makes a crucial point in your paper.

p. 8, lines 376-377.  You might want to remind readers that the administrators in this study removed the target from the classroom, instead of the bullies.

p. 9, lines 390-392.  Consider revising for clarity to: “…”undue burdens”, referring to the often invisible work imposed on disabled people and their families to ensure their participation.

p. 9, line 424.  Might be clarified if changed to “how young disabled people are made to feel out of place through the ways…”

p. 10, lines 438-440. Please revise this sentence to provide greater clarity.

Comments on the Quality of English Language

Generally, the English language in this article was carefully and clearly written.  There were some sections where clarification is necessary, and those are indicated to the authors in the review.  Please note the comments about the citation of the legislation that appears on lines 125-127.  

Author Response

Dear reviewer 1,

We sincerely thank you for your thoughtful comments and efforts to help us improve our manuscript. Below we list how we have responded to the specific points made (see comments in regular text and our answers in italics and bold. In the manuscript we have highlighted all major changes in yellow. Please note that not all changes may be fully visible since we cut out and synthesized parts of the text in line with recommendations.

Please remove the designation of sections (Aims, Methods, Findings, Conclusions) as they distract from your brief description.

This has been done

Please indicate what you were observing.

To better clarify, we have changed the phrasing into participant observation, as we are referring to the data-gathering method we used.

  1. 1, line 14. You discuss the enactment of young people’s rights as due to chance.  In your description, you seem to make the case that there were strong influences that shaped their experiences.

Many thanks for pointing out this controversy. We have added a sentence that better explains the many influences which shape participant´s experiences.

 Please edit this complex statement to provide greater clarity.

We have edited this sentence and believe it is more clear now. 

Please define “ableist assumptions” earlier in the introduction.

This has been done

 It would be helpful for you to clarify your meaning of “mediating social inclusion.”

We removed the word mediating and added the word facilitating instead. We feel it better catches the meaning we are referring to. Later in the paper we go deeper into explaining the meaning of facilitating social inclusion.

  1. 3, lines 125-131. Please cite these sources in a briefer way if possible.

We agree that the legal citations are quite long, it is always a challenge for us in Iceland to translate legislation into other languages since Icelandic legal texts traditionally are lengthy. In this revised version we skipped the English translation in text but it can be found in the reference list.

  1. 3, line 140-141. What is the specific lack of knowledge you are referring to here?

To better explain our meaning we have changed into awareness of disability issues. 

  1. 4, line 160. Please indicate what type of participation is possible.

We believe this should be clear as the observations focused on what young people were doing at their homes and schools and on their possibilities for participation and their engagement and interactions with peers and adults. To better explicate we have added the phrasing „within these environments“.

  1. 5 lines 200-201. Please specify how many comments from how many participants.

The approximate number of comments has been added to the text.

  1. 5.lines 219-220. Were “other participants” always parents? Please indicate the roles played by the other signers.

We have clarified the roles of other participants.The teachers signed to interviews and observations at school.

  1. 8. Lines 363-365. Please consider expanding this section, which makes a crucial point in your paper.

We have expanded this section and added

  1. 10, lines 438-440. Please revise this sentence to provide greater clarity.

We have revised this sentence and believe it is clearer now.

Reviewer 2 Report

Comments and Suggestions for Authors

I appreciate the research topic and qualitative approach which is convenient for this kind of research. However, the data analyses is too brief and not well described in the report. Moreover, the methodology is not presented in details, and it seems to be questionably designed. It would be nice to read better description of the coding process, observation parameters and its analysis, etc. Finally, it's hard to capture some new findings in the discussion and conclusion... I recommend to improve the above mentioned sections (methodology, discussion, conclusion) before the publication of this manuscript.

Author Response

We acknowledge your concerns. In our revised manuscript we have addressed some limitations in the sections you mentioned. 

Reviewer 3 Report

Comments and Suggestions for Authors

Thank you for doing this study. I particularly appreciate the inclusivity of your research design to the people it affects (involving children with developmental disabilities directly with the research such as providing drafts for them to review and add comments), which is sadly lacking in much of research. I anticipate that this study will have significant impact on children and families in Iceland and also other Nordic countries with similar socio-cultural dynamics and community structures. 

I just want to double check one thing -- for two participants in Table 1, their condition is "hidden". Is that due to confidentiality reasons? If so, there should be a line that mentions it perhaps in a caption just under the table.

Author Response

Dear reviewer 3,

We sincerely thank you for your thoughtful comments and efforts to help us improve our manuscript, your kind words and compliments are truly appreciated. The hope is that the paper will ignite discussions and more awareness of young disabled people's issues and the legal environment. Hopefully, we will see some changes in the near future. 

To answer your question, when referring to hidden impairments, we are talking about impairments that are not necessarily immediately apparent. People who identify as having hidden disabilities belong to a diverse group who face varied challenges. Many have chronic illnesses and/or mental/psychological impairments. To clarify our meaning of the concept, we have changed the word hidden into invisible.      

In addition, we have made several minor changes, highlighted in yellow.

Round 2

Reviewer 1 Report

Comments and Suggestions for Authors

In its revised version, this article should be of substantial interest to the readers of Social Sciences, since it presents key ways in which young people affected by disabilities can be supported in both in their physical and social lives. As stated in my earlier review, the article is enriched by both the examination of their social situations, and their movement toward greater wellness.  Your revisions have resulted in a clearer statement of the situation of the young people that were participants in the study, and have resulted in a manuscript that is quite clear and compelling.  Just one minor change is suggested.  In line 453-454, please replace "human right perspective" with "human rights perspectives"